# Sugar Production from Hybrid Poplar Sawdust: Optimization of Enzymatic Hydrolysis and Wet Explosion Pretreatment

**DOI:** 10.3390/molecules25153396

**Published:** 2020-07-27

**Authors:** Rajib Biswas, Philip J. Teller, Muhammad U. Khan, Birgitte K. Ahring

**Affiliations:** 1Bioproducts, Sciences and Engineering Laboratory, Washington State University, Tri-Cities, 2710, Crimson Way, Richland, WA 99354, USA; r.biswas@tricity.wsu.edu (R.B.); pt@biovantage.dk (P.J.T.); muhammadusman.khan@wsu.edu (M.U.K.); 2Biological Systems Engineering, L.J. Smith Hall, Washington State University, Pullman, WA 99164, USA; 3The Gene and Linda Voiland School of Chemical Engineering and Bioengineering, Washington State University, Pullman, WA 99163, USA

**Keywords:** biorefineries, enzymatic hydrolysis, fermentable sugars, hybrid poplar, wet explosion pretreatment

## Abstract

Wet explosion pretreatment of hybrid poplar sawdust (PSD) for the production of fermentable sugar was carried out in the pilot-scale. The effects of pretreatment conditions, such as temperature (170–190 °C), oxygen dosage (0.5–7.5% of dry matter (DM), *w*/*w*), residence time (10–30 min), on cellulose and hemicellulose digestibility after enzymatic hydrolysis were ascertained with a central composite design of the experiment. Further, enzymatic hydrolysis was optimized in terms of temperature, pH, and a mixture of CTec2 and HTec2 enzymes (Novozymes). Predictive modeling showed that cellulose and hemicellulose digestibility of 75.1% and 83.1%, respectively, could be achieved with a pretreatment at 177 °C with 7.5% O_2_ and a retention time of 30 min. An increased cellulose digestibility of 87.1% ± 0.1 could be achieved by pretreating at 190 °C; however, the hemicellulose yield would be significantly reduced. It was evident that more severe conditions were required for maximal cellulose digestibility than that of hemicellulose digestibility and that an optimal sugar yield demanded a set of conditions, which overall resulted in the maximum sugar yield.

## 1. Introduction

The increasing focus on greenhouse gas emissions from fossil energy production, along with the decreasing resource of fossil fuels, has increased the global interest in finding alternative resources for the production of fuel, chemicals, and materials. Lignocellulosic materials are a promising feedstock for the production of industrial chemicals, materials, as well as biofuels. In the United States, more than one billion tons of lignocellulosic biomass can be produced annually, which can be used for the production of biofuels to replace the domestic use of fossil fuels and for substituting the use of fossil fuel for producing chemicals [1]. Currently, about 370 million tons of forest residue is produced for the production of paper, energy, and other products [2]. Among the lignocellulosic biomass materials, hybrid poplar (*Populus* spp.) is considered a promising feedstock for the production of cellulosic biofuels due to its large genetic diversity, availability for harvest over the whole year, fast growth rate, as well as higher bulk density as compared to the herbaceous feedstock, which eases the transport and storage [3].

Hybrid poplar (*Populus* spp.) is being cultivated largely in the Pacific Northwest region of the United States and has emerged into commercial production, occupying about 50,000 acres [4]. The aboveground biomass yield of a 4 years old hybrid poplar is 11 MgHa^−1^year^−1^ [5]. Due to its large availability, fine tissue, and lower cost, this wood is widely being used for the timber and furniture industry [6]. During the processing of this hardwood into products, a large amount of sawdust is produced [7]. This residue could potentially be a cheap and abundant feedstock for the production of biofuels. However, very few studies have been done on the use of hybrid poplar sawdust as a feedstock for biofuel production along with the potential for producing cellulosic sugars from this raw material. This material contains a mixture of bark and left-over woody materials after removal of the inner woody core for producing the specific bio-products.

Poplar sawdust (PSD) could potentially be an ideal substrate for the conversion of its carbohydrates into intermediate fermentable sugars and then for the subsequent production of fuels and chemicals, using microbial processes. This is partly due to its reduced particle size and high content of carbohydrates, such as cellulose and hemicellulose (~70%), along with low ash content. Moreover, this feedstock is widely available at a cheaper cost and will not, in any way, compete with human or animal consumption [8,9]. However, as compared to the industrial technologies developed for the conversion of sucrose and starch-rich materials to biofuels, a pretreatment step is necessary for this type of feedstock to achieve efficient hydrolysis of the carbohydrates into fermentable sugars. During the pretreatment, the rigid structure found in the cell wall is disrupted, and the major polymers, such as cellulose, hemicellulose, and lignin, can be accessed by cellulolytic enzymes [10,11,12,13,14]. The content of lignin and the degree of crystallinity of cellulose are major factors, which determine the efficiency of the enzymatic hydrolysis of lignocellulosic material [15,16]. Therefore, different types of pretreatments, such as alkali pretreatment [16], hydrothermal treatment [17], steam explosion pretreatment [18], dilute acid pretreatment [19], irradiation pretreatment [20], wet explosion [21], organic solvent pretreatment [22], hot compressed water pretreatment [23], and ionic liquids (ILs) pretreatment [24], have been used for improving enzymatic hydrolysis. However, different pretreatments have different advantages and disadvantages, such as risks of environmental pollution, high cost of chemicals added, lower efficiency, and strict operating parameters, which not only makes the pretreatment process costly but is further reducing the feasibility of the pretreatment process to be commercialized [18].

Recently, the wet explosion pretreatment method [25,26] has been advanced to pilot scale [27], where it was shown to outperform existing thermochemical pretreatment methods for processing softwood forest residues [27] and other agricultural residues [21,28]. In the wet explosion (WEx) method, biomass is exposed to an oxidizing agent (e.g., pure oxygen, air, or H_2_O_2_) under high temperature and pressure for a total of 20 to 30 min. Besides its effectiveness in increasing carbohydrate hydrolysis, this method offers additional advantages, such as minimal water use, no requirement for chemical recovery/detoxification, the minimum formation of degradation products, and higher lignin solubility [10,21,28,29,30]. Currently, hemicellulose is being fractionated by using dilute acids, bases, or enzymatic hydrolysis. But the cost of enzymatic hydrolysis limits the production of sugars from hemicellulose at the industrial scale. Besides, this additional separation step is required for the removal of the acid catalysts before the utilization of sugar monomers. However, the wet explosion pretreatment can be used for selective hydrolysis of the hemicellulose fraction of biomass. High solids concentration is beneficial for lowering the cost of pretreatment and for increasing the sugar yield after enzymatic hydrolysis. The wet explosion pretreatment process can be operated at high solids level, and enzyme hydrolysis can further be done with higher solids level of pretreated material compared to several other pretreatment methods due to the low concentrations of inhibitors developed during this pretreatment [31]. Overall, the efficacy of pretreatment is largely influenced by biomass species.

The wet explosion pretreatment has been studied widely for agricultural residues and softwoods, where it has shown promising results, but, previously, it has not been studied for hardwood. Therefore, this study is the first one where hardwood is used. The process optimization is done for maximizing the formations of cellulosic sugars from this material. The optimization will further focus on reducing the formation of various aldehydes formed by the degradation of monomeric sugars during the pretreatment process. The formation of these inhibitory products not only reduces the yield of sugars during pretreatment, but it also hinders the bacterial and yeast fermentation process in concentration as low as 0.25% *w*/*w* [32].

In the present study, the process parameters of WEx pretreatment was tested in a pilot-scale setup during the pretreatment of poplar sawdust. The central composite design methodology was used to design a series of experiments to evaluate the effects of the process parameters on enzymatic saccharification of pretreated poplar sawdust (PPSD). The pretreatment conditions, including temperature, the dosage of oxygen, and residence time, were statistically analyzed to identify the optimal combination of parameters by evaluating responses using response surface methodology. Further, the conditions, such as temperature and pH, for successful enzymatic saccharification of the pretreated material with commercial enzymes were optimized. The surface hydrophobic properties of the lignocellulosic biomass slurry differ from pure cellulose, which is typically used commercially for activity testing.

## 2. Results and Discussion

### 2.1. Composition of Poplar Sawdust

The chemical composition of poplar sawdust (PSD) is shown in Table 1. The total carbohydrate content of PSD accounts for 60.9%, consists of glucan, xylan, galactan, arabinan, and mannan of 41.8%, 14.9%, 0.9%, 0.6%, and 2.7% of the total weight (oven-dry basis), respectively. Further, PSD consists of a 3.6% acetyl group and 31.9% total lignin (acid-soluble and insoluble) on an oven-dry basis. Extractives in PSD measured in sequential extraction with both water and ethanol in which the extractives were found to be 2.1% and 3.9%, respectively. While glucan and xylan contents in the poplar sawdust were found to be lower than that of milled biomass of whole hybrid poplar, as previously reported [33], the lignin content of the poplar sawdust residues was found to be higher. This could be due to the content of bark residues with higher lignin content in the poplar sawdust sample used in this study.

### 2.2. Effect of Wet Explosion Pretreatment on Poplar Sawdust

The optimization of wet explosion (WEx) pretreatment of poplar sawdust for subsequent enzymatic hydrolysis was based on the experimental design of 17 pretreatment runs (Table 2) performed in pilot-scale, with an initial dry-solid concentration of 30%. The process parameters, such as temperature (T, 170–190 °C), oxygen dosage ([O_2_], 0.5–7.5% of dry matter (DM)), and residence time (t, 10–30 min), were treated as factors to design the experiments. To assess process variability, three runs (8, 9, and 10) were executed at the central point of process parameters, i.e., T = 180 °C, [O_2_] = 4% of DM, and t = 20 min. The mean value and standard deviation of the DM content of the PSD for the central runs were 28.3% ± 1.3. DM content after the pretreatments appeared to be lower in all the pretreatments, except for the runs 3, 4, 7, and 13, where DM contents were slightly above 30%.

#### 2.2.1. Composition of the Liquid Fraction

The composition of the liquid fraction of PSD after wet explosion pretreatment was one of the parameters examined in this study to evaluate the effects of process parameters. After the pretreatments at a high solid concentration (30% DM), the PSD hydrolyzates resulted in a liquid containing a mixture of compounds, including monomers (glucose, xylose, galactose, arabinose, and mannose), oligomers, organic acids (e.g., acetic acid), and furans, such as hydroxymethylfurfural (HMF) and furfural. The dilute acid hydrolysis of the liquid fractions was performed to hydrolyze the oligomers into monomers and to quantify the total sugars as monomers using HPLC analysis. The concentrations of total monomeric sugars after dilute acid hydrolysis of the liquid fraction obtained from the WEx pretreatment of PSD under various conditions are shown in Table 2. Further, the degradation products as generated after pretreatment and found in the liquid phase are also reported in Table 2.

Maximum solubilization of total carbohydrate occurred in the pretreatments carried out at median temperatures (T = 180 °C). The effects of other parameters on the solubilization of carbohydrates were determined by quantifying the total sugar concentration in the liquid fraction right after the pretreatment, as shown in Figure 1, at a temperature of 180 °C. As displayed in the response surface plot (Figure 1), the highest concentration of total sugars in the liquid fraction could be achieved with an oxygen loading of above 4%. According to the modeled data, a residence time greater than 22 min had an adverse effect on the total sugar concentration of the liquid fraction, and a reduction of the overall sugar concentration was observed. This could primarily be due to the degradation of sugars into other products during longer residence times. Interestingly, the hemicellulosic sugars are found mostly in the oligomeric form in the liquid fraction; for example, the concentration of xylooligosaccharides was 2–3 times higher than for the xylose monomer. The release of hemicellulosic sugars in the form of oligomers contributes to the reduction of sugar degradation products, such as organic acids, HMF, and furfural [34,35]. Both cyclic and acyclic mechanisms involved in the degradation of glucose and xylose into HMF and furfural, respectively, are enhanced by the availability of free mono-sugar. Consequently, the formation of most degradation products, such as acetic acid, HMF, and furfural, in traditional dilute acid pretreatment is multiplied by several folds when compared with hydrothermal pretreatment [36]. The composition of the aforementioned degradation products found in the liquid fraction of PSD is shown in Table 2. As observed, acetyl groups from the side chain of hemicellulose were the first to be hydrolyzed at elevated temperatures, followed by arabinan and xylan. Further, an increase in acetic acid concentration in the reaction under such conditions causes further autohydrolysis of the lignocellulosic materials and improve the overall effect of temperature and residence time on biomass structural degradation [37].

In WEx, the provision of an exothermal process at temperatures above 170 °C due to the addition of oxygen not only reduced the heating requirements [26,37] but was also found to be effective in minimizing the formation of degradation products through better process efficiency at lower process severity. As expected, the highest concentrations of acetic acid, HMF, and furfural of 8.3, 0.9, and 2.2 g/100 g (oven-dry basis), respectively, were found in the pretreatment carried out at the harshest conditions applied in this study (T = 190 °C, t = 30 min, and [O_2_] = 7.5% of DM). Although these degradation products are known to be inhibitory at higher concentrations for subsequent biological conversion processes [36,38], at lower concentrations, these products can be metabolized and used as a carbon source for microorganisms [39]. Interestingly, the concentration of potential inhibitors was found even under the harshest conditions applied in our study to be within the acceptable range [32] for upstream biological processes as well as downstream processing.

#### 2.2.2. Enzymatic Hydrolysis of Pretreated Poplar Sawdust

The pretreated poplar sawdust samples were digested with a mixture of enzymes (Cellic^®^ CTec2 and Cellic^®^ HTec2 kindly provided by Novozymes, Franklinton, NC, USA) at initial enzyme dosages of 16.7 mg enzyme proteins (EP)/g DM, of which 11.7 mg EP from CTec2 (mainly cellulolytic activity) and 5.0 mg EP from HTec2 (mainly hemicellulolytic activity), were applied for all the pretreated samples and monitored for cellulose digestibility. These commercial enzymes are capable of hydrolyzing the carbohydrate polymers into monomeric solutions, while the non-digestible compounds, such as lignin, remain as an insoluble residue. The efficiency of the pretreatment for lignocellulosic biomass can be evaluated based on the sugar yields after enzymatic hydrolysis [21,33]. The cellulose digestibility after the enzymatic hydrolysis of pretreated PSD at different residence times (10–30 min) was calculated from the percentage of glucan obtained after the hydrolysis of original glucan input [21]. In the initial screening of pretreated samples, the highest glucan yield of 87.1% ± 0.1 was obtained under pretreatment performed at 190 °C with 30 min residence time and 7.5% O_2_ of DM (run#17). Figure 2 depicts the response surface model of cellulose digestibility after the enzymatic hydrolysis of pretreated PSD. All the parameters tested in this study had an influence on the glucan yield, while the temperature was the key factor. It is apparent that the impact of temperature on cellulose digestibility increases with the increase in the concentration of O_2_ regardless of the residence time. With the same temperature of 190 °C, cellulose digestibility increased from 56.5% ± 0 in run 13 to 87.1% ± 0.1 in run 17, with an increase in O_2_ concentration from 0.5% to 7.5% at 30 min residence time. However, the hemicellulose digestibility/yield was found to be significantly lower in run 17 (32.7% ± 0). This could be due to the fact that increasing the pretreatment severity would result in increased degradation of hemicellulosic sugar. On the other hand, the maximum hemicellulose digestibility of 90.8% ± 0 was achieved in run 5 with a temperature of 170 °C, 7.5% O_2_ concentration, and 30 min residence time. The results for each response of cellulose digestibility and hemicellulose digestibility with the predicted *p*-values (*p* < 0.0001) and the coefficient of determination (R^2^) are depicted with the fitted correlation of actual and predicted values in Figure 3. The prediction model for both cellulose digestibility and hemicellulose digestibility after enzymatic hydrolysis showed a satisfying correlation coefficient (R^2^ = 0.97). The maximum desirability (0.76) in the predictive model showed a cellulose digestibility of 75.1% and hemicellulose digestibility of 83.1%, which were achieved at a temperature of 177 °C with 7.5% O_2_ concentration and 30 min residence time. Although this cellulose digestibility (75.1%) was lower than the values obtained in optimal conditions for cellulose hydrolysis, nonetheless, the cellulose digestibility (75.1%) found in this study was still higher than the values obtained by Kim et al. [40] on hybrid poplar using liquid hot water pretreatment at 200 °C for 10 min at 15% solids. In their studies [40], total monomeric sugar yield (glucose and xylose) reached 67% after 72 h of hydrolysis when 40 Filter Paper Units (FPU) cellulase per gram glucan was used, which is significantly higher enzyme loadings compared to our study. With 15 FPU cellulase per gram glucan, the glucose yield was only 54% after 120 h of enzymatic hydrolysis [40]. The results of cellulose and hemicellulose digestibility are comparable with other lignocellulosic feedstocks pretreated by wet explosion pretreatment, as shown in Table 3.

#### 2.2.3. Enzymatic Hydrolysis Optimization

The enzymatic hydrolysis conditions, such as temperature, pH, and CTec2/HTec2 ratio, were screened to determine the effects of the parameters on glucose yields. Typically, pure cellulose is used as a substrate to evaluate the optimal conditions for enzymatic saccharification. However, the surface hydrophobic properties of pure cellulose differ significantly from lignocellulosic biomass, especially after lignocellulosic biomass undergoes thermochemical pretreatment under elevated temperatures. It was also previously shown that pH has an effect on the amounts of cellulases, which are bound productively to cellulose, making up a fraction that does not participate in cellulose hydrolysis due to nonproductive attachment with lignin [44]. PPSD sample obtained in run 17 was digested with a mixture of enzymes, (Cellic^®^ CTec2 and Cellic^®^ HTec2) with 3 different ratios (4:0, 3:1, and 2:2) at a total of 8.4 mg EP/g DM, was tested in a temperature range between 40 °C and 55 °C and a pH range between 4.5 and 5.5, and incubated for 72 h. The low enzyme dosage was chosen to closely observe the effect of the variable conditions applied. As shown in Figure 4, the maximum cellulose digestibility could be achieved using the mixture of enzymes of CTec2 and HTec2 with a ratio of 3:1 under the conditions of pH 5.5 and hydrolysis temperature of 47.4 °C. According to the statistical analysis, the desirability of 0.973 could be achieved using the set of conditions for enzymatic hydrolysis. Overall, using this optimal set of conditions for enzymatic hydrolysis, the cellulose digestibility of pretreated poplar sawdust could be further enhanced by 8.4% on top of the achieved sugar yields while using the recommended conditions from the enzyme provider.

## 3. Materials and Methods

### 3.1. Raw Materials

Hybrid poplar (*Populus* spp.) sawdust (PSD) used in the study was obtained from Upper Columbia Mill (UCM) in Boardman, OR, USA, processing lumber products. The original poplar sawdust dry matter (DM) was determined to be 95.8% (volatile solids = 99.1% of total solids (TS); ash = 0.9%). The poplar sawdust was powdery with a particle size of 2–4 mm and was used as obtained without further size reduction. The chemical composition is shown in Table 1.

### 3.2. Compositional Analysis

The moisture, carbohydrates, acid-insoluble lignin (Klason lignin), acid-soluble lignin, and ash contents of original raw materials were determined by analytical procedures developed by the National Renewable Energy Laboratory (NREL) [45,46]. The first stage hydrolysis of 0.3 g samples of biomass was performed with 3.0 mL of 72% (*w*/*w*) H_2_SO_4_ for 1 h at 30 °C in a water bath. The 84 mL of deionized water (DI) was added to dilute the hydrolyzates to 4% H_2_SO_4_ (*w*/*w*) concentration and autoclaved at 121 °C for 1 h. Sugar monomers were analyzed on an Aminex HPX-87H column (Bio-Rad, Hercules, CA, USA) at 60 °C with 4 mmol H_2_SO_4_ as eluent with a flow rate of 0.6 mL/min. Composition analysis was carried out in triplicates, and sugar contents were quantified by comparison with sugar calibration standards.

### 3.3. Wet Explosion Pretreatment

The wet explosion pretreatment (WEx) of poplar sawdust was performed in a pilot-scale set-up that includes a 100 L stainless steel pressure reactor equipped with an anchor-shaped scraper type mixer. The target temperature was reached by direct steam injection, replacing residual air in the headspace. The mixing speed of 50 rpm was maintained during the heating period (approx. 10 min). The final temperature was maintained with an external oil heater until the reaction was terminated. Oxygen was purged into the reactor, and an elevated mixing speed of 75 rpm was applied. After the predetermined residence time, the reaction was terminated by opening the discharge valve into a 250 L flash tank, initiating a sudden decompression of the materials in the reactor. In each run, the 100 L pretreatment reactor was impregnated with 4 kg (oven-dry basis) of poplar sawdust, and tap water was added to reach a solid concentration of 30%. The pretreatment conditions applied for the series of runs are shown in Table 2. A sample was collected to carry out liquid analysis for sugars, organic acids, and furans. Enzymatic hydrolysis experiments were conducted on pretreated slurry without any solid–liquid separation to minimize process steps.

### 3.4. Design of Experiment

The wet explosion pretreatments of PSD were performed at temperatures ranged from 170–190 °C based on the preliminary experiments and available literature data [21,27]. Oxygen dosage used in this study varied in the range of 0.5–7.5% of DM (dry matter, *w*/*w*), with a residence time of 10–30 min. A central composite design (Table 2), including three central points of the factors, was used to design the experiments. A series of 17 runs with different operational conditions was suggested in order to generate regression reports of the factors and responses using response surface methodology. The yields of glucan, xylan, and total carbohydrate after the enzymatic hydrolysis of the pretreated samples were determined to evaluate the optimal set of parameters for the pretreatment. In addition, monomeric and oligomeric sugars, acetic acid, and furans (5-hydroxymethylfurfural and furfural) were further evaluated.

### 3.5. Analysis of Liquid Fraction after Pretreatment

The liquid fraction of pretreated poplar sawdust (PPSD) from each pretreatment run was analyzed for monosaccharides (glucose, xylose, and arabinose), acetic acid, 5-hydroxymethylfurfural (HMF), and furfural by an HPLC system equipped with an Aminex HPX-87H column (Bio-Rad, Hercules, CA, USA), as described in compositional analysis subsection. The acid hydrolysis of the liquid was carried in Eppendorf tubes using 4% H_2_SO_4_ (*w*/*w*) at 99 °C for 4 h and neutralized using an equivalent amount of barium hydroxide. Dilute acid hydrolysis of the liquid samples was carried out in triplicates, and the samples were always filtered using a 0.45 µm filter (PTFE membrane, Acrodisc^®^ syringe filters, 13 mm, Pall^®^ Life Sciences, Pensacola, FL, USA) before chromatographic analysis.

### 3.6. Enzymatic Hydrolysis

Enzymatic hydrolysis of PPSD samples was performed with a mixture of Cellic^®^ CTec2 and Cellic^®^ HTec2 (Novozymes, Franklinton, NC, USA). Initial enzyme dosages of 16.7 mg enzyme proteins (EP)/g DM, of which 11.7 mg EP from CTec2 and 5.0 mg EP from HTec2, were applied for all the pretreated samples. The enzyme proteins of CTec2 and HTec2 were determined to be 265 and 235 mg EP/mL, respectively, using Pierce^®^ BCA^®^ protein assay kit following the company’s procedure (Thermo-Fisher Scientific, Rockford, IL, USA). Five grams of oven-dry substrate were supplemented with 5 mL 1 M citrate buffer (pH = 5.0) and aforementioned enzyme mixtures and then added into DI water to achieve a 5% (*w*/*w*) consistency of solution. One milliliter of sodium azide solution (2%, *w*/*w*) was used in the media to inhibit microbial contamination. However, 5% consistency of solution with enzyme dosages of a total of only 8.4 mg EP/g DM was used in order to optimize the enzymatic hydrolysis conditions, such as pH, temperature, and enzyme mixtures, incubated at 50 °C. Enzymatic hydrolysis optimization experiments were carried out at 40/45/50/55 °C, according to experimental design, in an incubator shaker (the lab companion IS-971 [R/RF] floor model incubated shaker, Jeiotech Co. Ltd., Daejeon, Korea) at 180 rpm for 72 h. After the enzymatic hydrolysis, the hydrolyzate sample was centrifuged at relative centrifugal force (RCF) of 20,817× *g* for 10 min at 4 °C and filtered (0.45 mm) for HPLC analysis. All the experiments were performed in duplicate using 250 mL screw cap glass vial with an active volume of 100 mL. Reported values were corrected for the sugar contribution from the enzyme mixture, as found in the blanks.

### 3.7. Statistical Data Analysis

The design of experiments and the experimental data were analyzed using JMP Pro statistical software (SAS, version 11.0). Total solubilized carbohydrates, organic acids, HMF, and furfural in liquid fraction and as post-pretreatment responses were assessed. Further, the yields of predominant sugars—glucan and xylan—in terms of cellulose and hemicellulose digestibility after the enzymatic hydrolysis were evaluated as responses of the examined factors (temperature, time, and oxygen dosage). The analysis was carried out in a fitted correlation using second-degree interaction under response surface macros and standard least-square personality with an emphasis on effect leverage.

## 4. Conclusions

This was the first time that WEx was used to pretreat poplar sawdust. In this study, we optimized the wet explosion pretreatment conditions for poplar sawdust for the optimum production of fermentable sugars. After the enzymatic hydrolysis, the highest total sugar yields (from the conversion of 75.1% cellulosic and 83.1% hemicellulosic) could be achieved at pretreatment conditions of 177 °C with 7.5% O_2_ and 30 min residence time. However, cellulosic sugar yield increased with up to 87.1% by using an operational temperature of 190 °C. However, the hemicellulosic sugar yield would decrease in these conditions compared to the use of lower temperatures, and sugar yield of 90.8% for the hemicellulose fraction could be achieved at 170 °C. Furthermore, the overall cellulose digestibility could be enhanced by 8.4% when using CTec2:HTec2 in a 3:1 relation at 47.4 °C and pH 5.5, which is different from the conditions described as optimal by Novozymes, the enzyme provider.

## Figures and Tables

**Figure 1 molecules-25-03396-f001:**
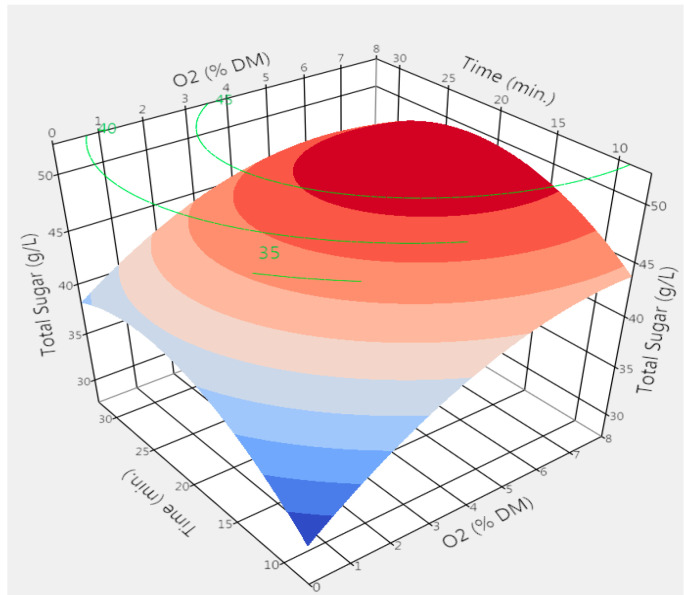
The concentration of total monomeric sugars after the dilute acid hydrolysis of the liquid fraction obtained after wet explosion pretreatment of poplar sawdust at median temperature (180 °C) at different O_2_ concentration and residence time.

**Figure 2 molecules-25-03396-f002:**
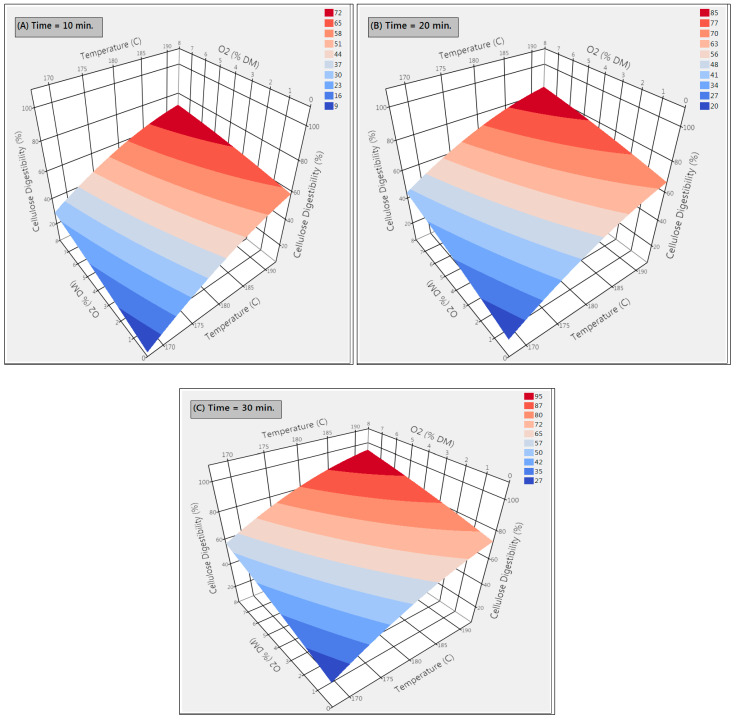
Response surface plots, showing the influence of temperature and residence time on cellulose digestibility (%) after enzymatic hydrolysis of pretreated poplar sawdust at different residence times; (**A**) residence time 10 min; (**B**) residence time 20 min; (**C**) residence time 30 min.

**Figure 3 molecules-25-03396-f003:**
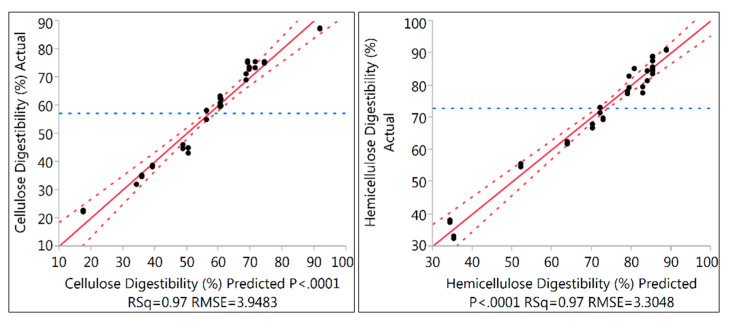
Fitted correlation of actual and predicted values, showing cellulose digestibility and hemicellulose digestibility with *p*-values and the coefficient of determination (R^2^). The maximum desirability in the predictive model is showing the cellulose digestibility and hemicellulose digestibility under different pretreatment conditions and with an optimal set of conditions.

**Figure 4 molecules-25-03396-f004:**
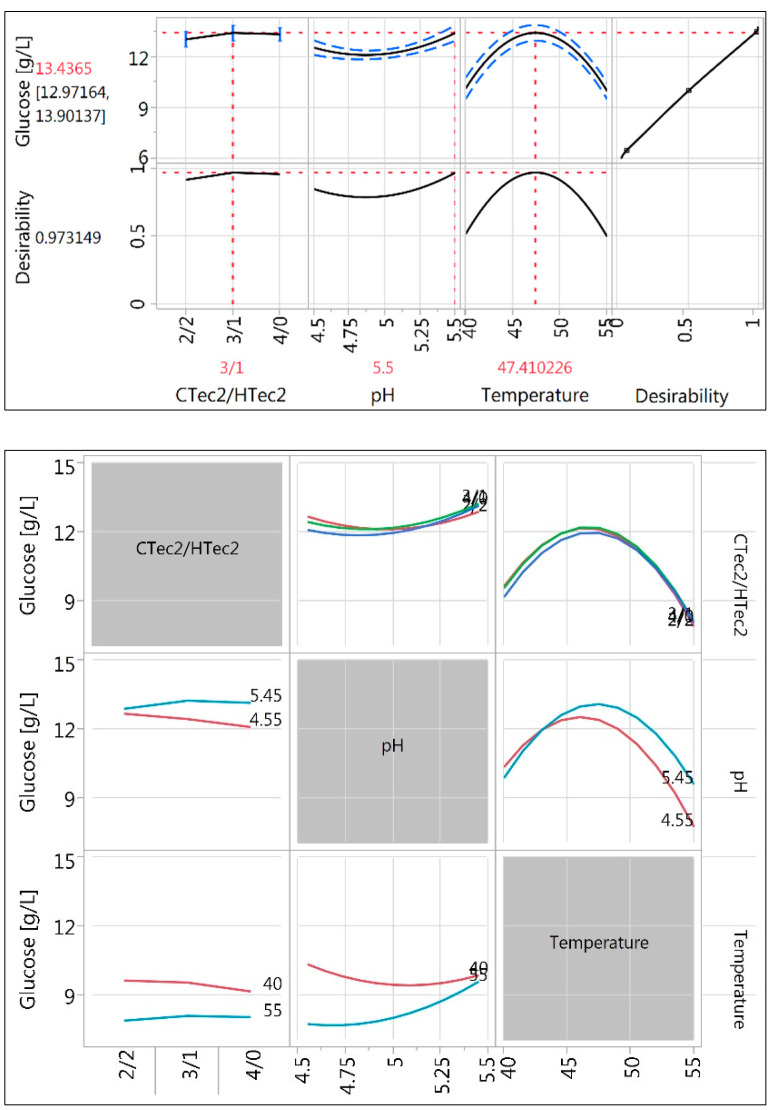
The maximum desirability, showing glucose concentration at the specific set of conditions for the enzymatic hydrolysis. The effects of individual variability of glucose concentration as a result of enzymatic hydrolysis under different conditions are shown.

**Table 1 molecules-25-03396-t001:** Chemical composition of poplar sawdust (PSD) raw biomass.

	G/100g Dry Matter *
Glucan	41.8
Xylan	14.9
Galactan	0.9
Arabinan	0.6
Mannan	2.7
Acetyl	3.6
Total lignin	31.9
Acid soluble lignin	4.2
Acid insoluble lignin	27.7
Total extractives	5.9
Water extractives	2.1
Ethanol extractives	3.9

* Dry matter of PSD (as received) = 95.8% of the total solids; volatile solids = 99.1%; ash = 0.9% (structural inorganics = 0.6%, soil = 0.3%).

**Table 2 molecules-25-03396-t002:** Conditions used for wet explosion (WEx) pretreatment at 30% initial dry matter. Soluble sugars and degradation products were measured in the liquid phase after dilute acid hydrolysis.

Run	Temp., °C	Time, min.	O_2_, % DM	% DM of after WEx	Glucose (g/L)	Xylose (g/L)	Galactose (g/L)	Arabinose (g/L)	Mannose (g/L)	Acetate (g/100 g DM)	HMF (g/100 g DM)	Furfural (g/100 g DM)
1	170	10	0.5	28.9	1.9	9.8	1.9	1.5	1.5	0.9	0	0
2	170	10	7.5	29.1	2.8	21.0	3.2	2.5	2.7	1.2	0	0
3	170	20	4.0	30.1	3.0	29.5	3.9	2.3	3.7	1.7	0	0.1
4	170	30	0.5	31.5	2.6	23.7	2.8	1.4	3.5	1.5	0	0.2
5	170	30	7.5	25.7	4.4	34.3	4.1	1.8	5.7	3.0	0	0.4
6	180	10	4.0	29.2	2.8	26.4	3.5	1.4	3.5	1.5	0	0.1
7	180	20	0.5	30.2	2.8	28.9	2.9	1.3	3.6	2.0	0	0.4
8 (central)	180	20	4.0	27.0	3.9	33.7	3.8	1.8	5.7	3.1	0.1	0.5
9 (central)	180	20	4.0	29.5	3.6	31.2	3.5	1.7	5.2	3.0	0.1	0.6
10 (central)	180	20	4.0	28.3	4.0	31.9	3.7	1.8	5.6	3.1	0.1	0.6
11	180	20	7.5	28.4	7.0	32.7	3.8	1.8	6.7	4.8	0.2	1.0
12	180	30	4.0	28.6	6.7	33.1	4.0	1.8	7.4	6.3	0.3	1.2
13	190	10	0.5	30.7	3.5	33.0	3.6	1.3	4.9	2.6	0.1	0.5
14	190	10	7.5	28.2	5.0	32.6	3.7	1.7	6.3	3.7	0.1	0.6
15	190	20	4.0	26.9	6.0	25.0	3.2	1.5	6.8	5.3	0.3	1.5
16	190	30	0.5	26.6	4.0	19.4	2.6	1.4	5.6	5.3	0.3	1.8
17	190	30	7.5	27.5	13.0	9.0	1.9	1.2	5.0	8.3	0.9	2.2

**Table 3 molecules-25-03396-t003:** Comparison of biomass digestibility with other feedstock pretreated by wet explosion pretreatment.

Feedstock	Pretreatment Conditions	Digestibility	Reference
Sugarcane bagasse	185 °C, 10 min, 16% of DM	87% cellulose	[21]
Loblolly pine	170 °C, 22 min, 25% of DM	96% cellulose and nearly, 100% hemicellulose	[31]
Wheat straw	180–185 °C, 15 min, 14% of DM	70% cellulose, 68% hemicellulose	[41]
Wheat straw	180–185 °C, 15 min, 14% of DM	69% cellulose, 55% hemicellulose	[41]
Miscanthus	170 °C, 5 min, 15% of DM	56% glucose, 32% xylose	[42]
Winter rye straw	195 °C, 15 min, 6% of DM	49% cellulose, 11% hemicellulose	[43]
Oilseed rape straw	195 °C, 15 min, 6% of DM	58% cellulose, 10% hemicellulose	[43]
Faba bean straw	195 °C, 15 min, 6% of DM	43% cellulose, 10% hemicellulose	[43]
hybrid poplar sawdust	177 °C, 30 min, 7.5% O_2_, 30% of DM	75.1% Cellulose, 83.1% hemicellulose	This study

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
