# Peer review of "Sugar Production from Hybrid Poplar Sawdust: Optimization of Enzymatic Hydrolysis and Wet Explosion Pretreatment"

_molecules, 2020, doi:10.3390/molecules25153396_

Round 1
Reviewer 1 Report
The manuscript - Optimization of Sugar Production from Hybrid Poplar Sawdust Using Wet Explosion Pretreatment by Biswas et. al., explores the efficiency of wet explosion pretreatment procedure to recover fermentable sugars from hardwood (hybrid polar) saw dust. The experimental design is appropriate and well summarized. Cellulose digestibility at 75% and hemicellulose at 83% is really excellent for hardwood. The paper, although is very well written, it could be further improved by addressing the following issues.
- In the introduction the authors state that “The Wet explosion pretreatment has been studied widely for agricultural residues and softwoods where it has shown promising results.” It would be nice to have table that compares the digestibility of various biomass to the present study.
- Define acronyms in Tables as footnote.
- Table 2 – The data is organized by temp in 1st column, time is 3rd column and O2, % DM in 2nd At any given temperature, time of wet explosion rather than O2 seem to make a significant difference. So, it is best switch these two columns.
- Further, for example wet explosion at 170oC - you can compare hexose and pentose content (cellulose and hemicellulose digestibility) recovery at 0.5% and 7.5% O2 at 10 and 30 min, but at 20 min % O2 is 4%, it is only done at 20 min and cannot be compared to anything else.? This is also true at other temperatures.
Author Response
We thank the Reviewers for their comments and have done the following revisions:
Reviewer 1
Comment 1
In the introduction, the authors state “The Wet explosion pretreatment has been studied widely for agricultural residues and softwoods where it has shown promising results.” It would be nice to have table that compares the digestibility of various biomass to the present study.
Our response: We have added the table in the revised manuscript
Comment 2
Define acronyms in Tables as footnote.
Our response: We have added Acronyms in the tables
Comment 3
Table 2 – The data is organized by temp in 1st column, time is 3rd column and O2, % DM in 2nd At any given temperature, time of wet explosion rather than O2 seem to make a significant difference. So, it is best switch these two columns
Our response: The columns have been switched in revised manuscript
Comment 4
Further, for example wet explosion at 170oC - you can compare hexose and pentose content (cellulose and hemicellulose digestibility) recovery at 0.5% and 7.5% O2 at 10 and 30 min, but at 20 min % O2 is 4%, it is only done at 20 min and cannot be compared to anything else.? This is also true at other temperatures
Our response: This is a result of the original experimental design, which does not affect the overall conclusions.
Yours sincerely,
Birgitte K. Ahring, Ph.D.
Professor, Head of BioScience and Technology Group
Bioproduct Sciences and Engineering Laboratory, BSEL
Washington State University
2710 Crimson Way
Richland, WA 99354-1671
Phone: 509-372-7682
Fax: 509-372-7690
Email: bka@wsu.edu

Reviewer 2 Report
The manuscript "Optimization of Sugar Production from Hybrid Poplar Sawdust Using Wet Explosion Pretreatment" by Biswas et al. presents experimental results on pretreatment procedure as it affects efficiency to recover fermentable sugars saw dust. The paper is well written, but please see my comments below:
- I think this paper that contains a lot of the concept of 'any' rather than 'why'.
- Since this is a first-time study, I think it is a very good direction to include various contents.
- 'Wet Explosion Pretreatment' was included in the title, but 'Enzymatic hydrolysis optimization' is also very important. I think author need to edit the title.
- Is there a separate table for 'Enzymatic hydrolysis optimization'?
- Except for Figure 1, it is difficult to see all the pictures. I am having difficulty reading the data axis and the exact values.
Author Response
We thank the reviewers for their comments and have done the following revisions:
Reviewer 2
Comment 1
I think this paper that contains a lot of the concept of 'any' rather than 'why
Our response
We do not really understand the comment. The paper is investigating the Wet Explosion pretreatment as a mean for treating hardwood, which has not been done before, and we believe it clearly shows that the concept has potential.
Comment 2
Since this is a first-time study, I think it is a very good direction to include various contents
Our response
We understand this comment as an acknowledgement of the method used to investigate the applicability of the pretreatment method.
Comment 3
Wet Explosion Pretreatment' was included in the title, but 'Enzymatic hydrolysis optimization' is also very important. I think author need to edit the title
Our response: We have revised the title
Comment 4
Is there a separate table for 'Enzymatic hydrolysis optimization'?
Our response: Even that we agree that both areas are of importance our focus is on the Wet explosion pretreatment while the enzymatic hydrolysis is less investigated in this paper. Basically we used modified hydrolysis conditions, which has already been described in the paper and we do not think a Table would add further to the paper.
Comment 5
Except for Figure 1, it is difficult to see all the pictures. I am having difficulty reading the data axis and the exact values
Our response: We have revised the Figures
Yours sincerely,
Birgitte K. Ahring, Ph.D.
Professor, Head of BioScience and Technology Group
Bioproduct Sciences and Engineering Laboratory, BSEL
Washington State University
2710 Crimson Way
Richland, WA 99354-1671
Phone: 509-372-7682
Fax: 509-372-7690
Email: bka@wsu.edu
